# The Management of Pediatric Acute Appendicitis—Survey among Pediatric Surgeons in Romania

**DOI:** 10.3390/medicina58121737

**Published:** 2022-11-27

**Authors:** Radu Bălănescu, Laura Bălănescu, Anna Maria Kadar, Tudor Enache, Andreea Moga

**Affiliations:** 1Pediatric Surgery Department, “Grigore Alexandrescu” Clinical Emergency Hospital for Children, 011743 Bucharest, Romania; 2Department of Pediatric Surgery and Orthopedics, “Carol Davila” University of Medicine and Pharmacy, 050474 Bucharest, Romania; 3Pediatric Surgery Department, “M.S. Curie” Clinical Emergency Hospital for Children, 077120 Bucharest, Romania

**Keywords:** appendicitis, children, survey, management

## Abstract

*Background and Objectives*: To assess the current practice pattern in the management of pediatric acute appendicitis in Romania. *Materials and Methods:* A questionnaire was emailed to all the members of the Romanian Society of Pediatric Surgery between June–July 2022. *Results*: 118 answers were received, 79.7% responses being from permanent staff members. In the diagnosis of appendicitis, complete blood count, C-reactive protein and abdominal ultrasound are the most commonly used diagnostic tools, while appendicitis scores are not widely used (25% of surgeons). In the case of simple appendicitis, 49.2% of surgeons prefer the conservative approach—oral/intravenous antibiotics. Those who choose the operative approach begin preoperative antibiotics in 56.7% of patients. In case of a stable patient, only 16.7% of surgeons will operate during the night. Laparoscopic approach is chosen by 51.7% of surgeons. In the case of a complicated appendicitis, 92.4% of surgeons will perform the appendectomy, prescribing preoperative antibiotics in 94% of the cases and continuing the therapy postoperatively in 98.2%. Laparoscopic approach is used by 28.8% of surgeons in case of complicated appendicitis. In presence of appendicular mass, 80% prefer a conservative approach with a delayed appendectomy within 6 months. Appendicular abscesses are managed operatively in 82.2% of the cases. The appendix is sent for histological analysis by 95.8% of surgeons. If the peritoneal cavity is contaminated, 95% of the respondents will take a sample for microbiological analysis, 71% will always place a drainage and 44% will always irrigate (71.9%-saline). *Conclusions:* Clearly, there seems to be a lack of consensus regarding several aspects of the management of acute appendicitis in children. In addition, minimally invasive surgery is not as widely used as reported, despite literature support.

## 1. Introduction

One of the most common complaints for pediatric patients presenting to the emergency department is abdominal pain. In most cases, this complaint is self-limiting, but nonetheless acute appendicitis should always be considered in the differential diagnosis [1].

Of all the abdominal surgeries, appendectomy is the most commonly performed in pediatric patients [2]. In the United States of America, around 70,000 children are diagnosed each year with acute appendicitis [3]. In Romania, due to a lack of data available, the incidence and prevalence of acute appendicitis in children could not be determined.

The management of appendicitis is a highly debated topic in spite of the great number of clinical guidelines and recommendations [4,5,6,7], due to the multiple presentations, diagnostic tools and surgical approaches.

Scoring systems for identifying cases of acute appendicitis have been devised such as the Alvarado score and the Pediatric Appendicitis Score (PAS), with initial trials showing a 76% sensitivity and a 72% specificity for the Alvarado score and a 100% sensitivity and 92% specificity for the PAS score. However, later studies found that none of these scores had an adequate predictive value and could not exclusively become the standard of diagnosis of acute appendicitis in pediatric patients. A new score was validated in 2008 by Anderson et al. that was developed to overcome the shortcomings of the Alvarado and the PAS score—the appendicitis inflammatory response (AIR). In an article published in 2021, Pogorelic et al. evaluated the validity of the AIR score and found that it has a high level, sensitivity and specificity and can be also used to differentiate perforated from non-perforated appendicitis in pediatric patients with a high level of accuracy [8,9,10,11].

Regarding the diagnostic process, in a literature review from 2016, Glass et al. found that in 65% of cases of children undergoing a surgical intervention for acute appendicitis an abdominal ultrasound was performed, while in only 42% was a computed tomography (CT)used [3].

Despite advances in the diagnosis of acute appendicitis, the misdiagnosis rate is still high, ranging from 28–57% in younger children and even more for those under the age of 2. Several possible non-invasive predictors were analyzed in order to help improve correct diagnosis such as white blood cell count, C reactive protein, hyponatremia, bilirubin levels and red cell distribution width [12,13,14].

Regarding the different management strategies, in a study published in 2020 by Barbora et al., 50 patients were randomized into two groups, with half of them receiving conservative treatment (anti-biotherapy) and the other half benefiting from surgical treatment, and after a 5-year follow up, the authors found that only 17% of the patients in the non-operative group required surgery for recurrent acute appendicitis.

As a tool for assessing the practice of different pediatric surgeons regarding appendicitis, several surveys were performed by Muelstedt et al. in 2004 [15] and Zani et al. in 2019 [16]. Questionnaires have a wide use in the field of medical research, regardless of whether patient or physician focused. Some examples include visual analogue scales, quality of life questionnaires, or scales rating the efficacy or safety of certain therapies [17].

Currently there is no definitive guideline for acute appendicitis in our country and we found the lack of an image of the current practice pattern regarding the management of this condition in pediatric patients. Therapy strategies greatly vary, not only depending on the institution, but also depending on the surgeon. We consider that a more homogenous approach to the treatment of acute appendicitis is necessary and for that reason we decided to assess the various management strategies preferred in our country. We decided to perform a survey among pediatric surgeons to assess the current practice pattern in the management of acute appendicitis in children, having as a second goal the comparison of our results with available data from the literature and the most recent survey, performed by Zani et al. [16].

We believe that this survey will be useful in starting the process of establishing guidelines for diagnosing, assessing and treating acute appendicitis in the pediatric population in Romania.

## 2. Materials and Methods

We designed a four-part questionnaire with questions regarding general information about the respondents, questions related to the diagnosis, perioperative management, surgical approach, postoperative management of pediatric patients with acute appendicitis and two extra questions regarding the use of an antibiotic Stewardship program in each hospital (Appendix A). The questionnaire included 34 questions which were initially sent for validation to the 11 members of the scientific committee of the Romanian Society of Pediatric Surgeons (RSPS). The members of the scientific committee were asked to evaluate each individual question rating them from 1 to 5 as either of no value or valuable for the study at hand. The data were analyzed using IBM SPSS statistics Data Editor ver. 29.0.0.0 (2022) using the Pearson’s Correlation Coefficient. Based on the score of the 11 members of the scientific committee, the questions were either kept in their initial form or modified according to the committee’s suggestions.

We then proceeded by emailing that questionnaire to all 154 members of the Romanian Society of Pediatric Surgery. Based on the number of members of the RSPS, we calculated that a minimum of 111 responses were required to have a 95% level of confidence that the sample size was statistically significant. The form was available for the members for a 6-week period (June–July 2022) and our study population consisted of pediatric surgeons who had returned a completed survey by the end of this 6-weeks period.

Data collection was performed using Google Form 2022 (Romania) and statistical analysis (frequency computing) was performed by using Excel ver. 16.27 (Microsoft Office Professional Plus 2013) and IBM SPSS statistics Data Editor ver. 29.0.0.0 (2022) Our present study was conducted under the approval of the local ethical committee. The aim of the study was to determine whether consensus existed in pre-, intra- and postoperative management of pediatric patients with acute appendicitis.

## 3. Results

### 3.1. General Aspects

In total, 118 complete answers were received over the 6-weeks period in which the questionnaire was available, marking a 76.6% response rate. Most of the respondents were permanent staff members (32 specialist, 62 consultants), while trainees account for 20.3%. Most of the surgeons work in a pediatric hospital (83; 70.3%). Most of the responders (36.4%) perform around 25–50 appendectomies per year (Table 1).

### 3.2. Preoperative Data—Work-Up

In order to establish the diagnosis of acute appendicitis, most of the surgeons will ask for a complete blood count, C reactive protein and an abdominal ultrasound, and 69.5% of surgeons responded that they rarely use a CT or an MRI in the diagnosis of acute appendicitis in pediatric patients (Table 2).

As an additional tool in the diagnosis of acute appendicitis some surgeons use either the Pediatric Appendicitis Score (PAS)—15.1% or the Alvarado score—15.1%, while the rest use neither. Out of the 30 respondents who use one or both scores, 6 (20%) were residents, while 24 (80%) were permanent staff members.

### 3.3. Operative Data—Simple Appendicitis

For a patient with simple, uncomplicated appendicitis, almost half of the responding surgeons would decide in favor of the operative approach (50.8%), while the others would prescribe oral (19%) or intravenous (81%) antibiotics, either on an outpatient basis (25.9%) or inpatient basis for a median of 5.76 days (IQR 5–7). When opting for a conservative approach, all the responding surgeons who recommended an oral course of antibiotics, prescribe cephalosporins. For those who opt for intravenous administration of antibiotics, cephalosporins are preferred in 70.2% of the cases, while in 29.8% of cases a beta-lactam or an association of antibiotics is used.

Overall, 83.3% of the surgeons who would prefer the operative approach, in case of a clinically stable patient, would not operate during the night. Regarding the use of antibiotics for an uncomplicated case of appendicitis, more than half of the respondents (56.7%) would choose to prescribe antibiotics preoperatively (cephalosporins in 82.4%; combinations of beta-lactams and macrolides in 17.6%).

In total, 85% of those who prefer the operative approach would most likely prescribe cephalosporines (80.4%) postoperatively (Table 3) for a median of 5 days (IQR 3–5), 11.7% would prescribe a single dose of intraoperative antibiotics (cephalosporins—71.4%; combinations of aminopenicillins, aminoglycosides and metronidazole—28.6%), while 3.3% would not prescribe any antibiotics.

Regarding the surgical approach, 61 of the surgeons who completed the questionnaire (51.7%) would prefer a minimally invasive approach, while the rest would operate using the open approach. When asked about the appendicular stump for the open approach, most of our respondents would invert the stump whenever possible (43.3%), 33.3% would let it free, 13.3% would always invert the stump, while 10% would leave it free whenever possible. In the laparoscopic approach, all the responding surgeons except one would use three trocars (one using four trocars), while for the appendicular stump the majority would use a knot (57.3%)—either intracorporeal (31.1%) or extracorporeal (26.2%). In addition, 37.7% of surgeons prefer to use the Endo-loop and 5% the stapler, and 56% of the respondents would use of endo-bag only in certain cases: 21% would always use it and 23% would never use it.

### 3.4. Operative Data—Complicated Appendicitis

Overall, 92.4% of the pediatric surgeons would perform appendectomy for a patient with complicated appendicitis, 6.8% would operate only in certain cases, while 0.8% would use a conservative approach, by administering intravenous carbapenems for 10 days. Of those surgeons who chose the operative approach, 60.7% would perform the surgery during the night if the patient was clinically stable, and 94% of surgeons recommend preoperative antibiotics, with cephalosporins being the antibiotic of choice in most cases (Table 4).

Postoperative antibiotics are prescribed by 98.2% of the surgeons, while only two surgeons responded that they would only prescribe antibiotics intraoperatively (cephalosporins). Cephalosporins and combinations of cephalosporins and other classes of antibiotics account for 65% of the responses (Table 5). The median period for postoperative antibiotic administration is 7 days (IQR 5–7).

Regarding the surgical approach, most of the surgeons (84; 71.2%) would prefer the open, while the others would perform minimally invasive surgery; 47.6% of those surgeons who prefer the open approach, would invert the appendicular stump whenever possible, 21.4% would let it free whenever possible, 20.25 would always let it fee, while 10.7% would always invert the stump. For the laparoscopic approach, all responding surgeons except one would use three trocars (the latter responded that they would use four trocars). As for the appendicular stump, most surgeons would use a knot—either intracorporeal (41.2%) or extracorporeal (20.6%); 32.4% of the surgeons who prefer the laparoscopic approach use the Endo-loop and 5.9% use a stapler; and while 70.6% of the pediatric surgeons would use the endo-bag only in certain cases, 11.8% would always use it, while 17.6% would never use it.

### 3.5. Appendicular Mass and Abscess

Around 80% of the respondents would treat a patient with an appendicular mass conservatively. The antibiotics the surgeons usually prescribe are represented in Table 6. In the event of a favorable outcome following an antibiotic course, 77% of surgeons would perform the appendectomy. The rest of the surgeons would perform the appendectomy only in certain cases (21%) and 2% would perform the appendectomy only at the parents’ request. Most of the surgeons would perform the appendectomy after 6 months (37.2%) with a wide range interval being reported (Table 7).

More than 80% of the surgeons would perform an appendectomy for a patient with an appendicular abscess, 11.9% would perform a percutaneous drainage (either ultrasound or CT guided), while 5.9% would prefer another approach.

### 3.6. Postoperative Data—Simple Appendicitis

In total, 61% of the respondents would discontinue antibiotic treatment after discharge, while the rest would prescribe an antibiotic course for a median of 5.41 days (IQR 5–7). Cephalosporins are the most commonly used (73.9%), followed by a combination of amoxicillin/clavulanic acid (17.4%) and aminopenicillins (4.3%). For patients with uncomplicated appendicitis, most surgeons would recommend postoperative follow-up for 7 days (46.6%) and 1 month (44.1%) after surgery.

### 3.7. Postoperative Data—Complicated Appendicitis

For patients with complicated appendicitis, 64.4% of the surgeons would continue to prescribe oral antibiotic according to Table 8 at discharge for a median of 6.16 days (IQR 5–7), with a minimum of 3 days and a maximum of 10 days.

For these patients, 58.5% of the surgeons would recommend postoperative follow- for a month, 20.3% for 6 months, while 18.6% of the surgeons would follow the patients for 7 days.

### 3.8. Varia

The appendix is sent for histological analysis by most of the surgeons—95.8%. If there is contamination of the peritoneal cavity, a sample is sent for microbiological analysis by 95% of the surgeons and a peritoneal drainage is be placed by 71%, whereas 1.7% of the surgeons would never drain the peritoneal cavity for a patient with purulent peritoneal fluid, while 5% would drain it only in certain cases.

As for the irrigation of the peritoneal cavity, 44% of the surgeons would do it only if there was contamination of the cavity, either with saline (71.9%), saline and antibiotics (10.5%) or other solutions (17.5%); 52% would always irrigate the peritoneal cavity, while 4% would never irrigate.

In most of the hospitals where our respondents work, there is no antibiotic stewardship program (82.4%), while 53.8% of the respondents have established anti-biotherapy protocols for the treatment of appendicitis.

## 4. Discussion

Our current survey comprises answers from permanent staff members and trainees, most of them (40.7%) performing more than 50 appendectomies per year.

Complete blood count, C reactive protein and abdominal ultrasound are frequently used for the diagnosis of suspected appendicitis, as other surveys report [16], but compared to that conducted by Muehlstedt et al. [15] the use of abdominal ultrasound is higher in our study. To reduce the negative appendectomy rates, the use of imaging methods is important, with abdominal ultrasound being the most used such method [18]. The CT is less used due to radiation risk in children.

Appendicitis scores (i.e., Alvarado, Pediatric Appendicitis Score, AIR score) are not widely used by the responding physicians, but 66.7% of the surgeons still use one or two of the scores when diagnosing acute appendicitis, as opposed to only 20% of the trainees.

The most common class of antibiotics used by respondents irrespective of the form of appendicitis is cephalosporins, either by itself or in various combinations. Taking into consideration the local bacterial epidemiology, it is difficult to perform a well sustained comparison to other studies.

Unsurprisingly, we found that almost half of the surgeons, in a case of simple appendicitis, would prefer a non-operative approach, prescribing either oral or intravenous antibiotics for 5 days, cephalosporins being the class of choice (oral 100%; intravenous 70.2%). There is growing evidence regarding this approach in the literature [19,20,21,22], but back in 2019, in the survey of Zani et al., only 15% of their respondents would have opted for the conservative approach.

According to the literature, most of the patients are given a single dose of preoperative antibiotics [16,23]. There are also recommendations for preoperative broad spectrum anti-biotherapy in children with non-perforated appendicitis [24]. In our survey, in a case of simple appendicitis, 56% of the surgeons would prescribe preoperative antibiotics and most (85%) would prescribe postoperative antibiotics for 5 days. As for cases of complicated appendicitis, preoperative anti-biotherapy is initiated by 94% of the surgeons—the same percentage as in the Zani et al. survey—and 98.2% of our respondents would continue the antibiotics for 7 days after surgery.

Cephalosporins are prescribed by 82.4% of our surgeons as a single dose preoperatively for a patient with simple appendicitis, while only 57.2% would recommend the same antibiotics for a patient with complicated appendicitis. For patients with complicated appendicitis, 30% of our respondents would prescribe a combination of cephalosporines and other class(es). A postoperative antibiotics course is frequently recommended in both simple and complicated appendicitis (85% vs. 98.2%), but the antibiotic choices is different. Cephalosporines alone are more frequently prescribed in simple appendicitis (80.39% vs. 40%), while combinations of cephalosporines and other classes are more frequently used for cases of complicated appendicitis (25% vs. 15.69%).

However, in the era of antibiotic resistance, the choice of antibiotics can be difficult and hard to standardize. Different antibiotic resistances have been reported depending on each country. For example, in a study conducted in China, Chan et al. reported in 2009 that they found a significant number of resistant bacteria, such as Pseudomonas aeruginosa and Escherichia coli [25]. In a study regarding antibiotic resistance in intra-abdominal infections, in Romania, Pricop et al. found in 2020 that 46% of intra-abdominal infections are due to aerobic microorganisms, while 54% resulted from the presence of anaerobic microorganisms. They also found that there is a rise in our country in the number of antibiotic resistant strains of anaerobic bacteria and, as such, surgeons should take this discovery into consideration when deciding which antibiotic therapy to use for their patients [26].

For a clinically stable patient with simple appendicitis, appendectomy would be performed during the night by 16.7% of the surgeons, while for the same scenario in a complicated appendicitis, the appendectomy would be performed by 60.7% of the surgeons. Compared to the most recent survey of pattern practice among EUPSA members [16], in our country the rate of appendectomy during the night is lower in cases of simple appendicitis (vs. 24%) but almost the same as that of complicated appendicitis (vs. 64%), all the results being in agreement with the literature [27,28,29].

Regarding the surgical approach, laparoscopy is preferred by half (51.7%) of the surgeons for cases of simple appendicitis, while for complicated appendicitis (except appendicular mass or abscess) laparoscopy is the preferred approach by almost one third of the surgeons (28.8%). It is known that the laparoscopic approach for appendicitis in children is well supported by a series of meta-analyses [30,31] and it should be the approach of choice in each center [16]. Concerning technical aspects, laparoscopic appendectomy is performed by almost all the surgeons using three ports, as other authors have reported [16]. Single-port appendectomy compared to the standard three ports approach seems to have the same outcome with higher surgery complexity and longer operative time [32]. During laparoscopic appendectomy either for simple or complicated appendicitis, resection of the appendix if performed by most of the surgeons using a knot (57.3% vs. 61.8%) rather than using an endo-loop (37.7% vs. 32.4%) or a stapler (5% vs. 5.7%). We were not able to assess the consequence of knotting on operative time, but Zani et al., reported a higher rate of endo-loop usage (73%) compared to suture ligation (9%). At the end of surgery, extraction of the appendix is done using an endo-bag by a similar percentage of surgeons (56% in simple, 70.6% in complicated appendicitis) as in the Zani et al. survey (64%) and in the same conditions, but only in certain cases.

Although there are a few studies concluding that routine peritoneal fluid cultures have no clinical usefulness [33,34], there is one such study showing that in a case of perforated appendicitis the cultures may be useful [35]. In our present study, a high number of surgeons (95%) send a sample for microbiological analysis if the peritoneal cavity if contaminated, and 71% of our surgeons will also place a peritoneal drain in case of contamination of the peritoneal cavity, a higher rate compared to the Zani et al. survey results in which, in the same scenario, only 5% of the surgeons would place a drain, while 52% would place a drain only in certain cases.

Irrigation of the peritoneal cavity is also a debated subject, with St Peter et al. [36] concluding that, during laparoscopic appendectomy in a case of perforated appendicitis, the irrigation of the peritoneal cavity compared to suction alone of the free fluid has no advantage. Even so, in our study, 52% of the respondents would always irrigate irrespective of the form of the appendicitis, usually using saline, and 44% would irrigate only if the peritoneal cavity was contaminated, as opposed to the data collected by Zani et al., where 29% of surgeons would always irrigate and 58% would irrigate only if a purulent collection was present.

In our study, the postoperative follow-up is longer in cases of complicated appendicitis (20.3% of surgeons would follow the patient for up to six months) compared to cases of simple appendicitis (5.1% surgeons would follow the patient for up to six months). Antibiotics are prescribed orally after the patient is discharged by more surgeons and for a longer period in case of complicated appendicitis compared to cases of simple appendicitis (64.4%; median 6.16 days vs. 39%; median 5.41 days), but in both cases cephalosporins are the most frequently prescribed (68.4% vs. 73.9%). In 2020, di Saverio et al. published an updated consensus for the diagnosis and treatment of acute appendicitis and recommended against using postoperative antibiotic therapy in pediatric patients with uncomplicated appendicitis and administering postoperative oral antibiotics for patients with complicated appendicitis for no longer than 7 days [37].

In special cases, such as an appendicular mass, 80% of our respondents would choose the conservative approach (vs. 75% in the Zani et al. survey) according to the literature and would always perform a delayed appendectomy in 77% of the cases (vs. 56% in Zani et al. survey), usually within 6 months after initial diagnosis. Appendicular abscesses are operated on by 82.2% of the surgeons, while in other surveys [16] the rate is lower, the majority preferring percutaneous drainage.

## 5. Study Limitations

An obvious drawback of our study is the fact that the questionnaire was distributed just to pediatric surgeons who are members of the Romanian Society of Pediatric Surgery, while there are probably pediatric surgeons who are not members of the Society, so our sample size was limited. In addition, amongst the answer we received were questionnaires filled by pediatric surgery residents and that can also be considered a limitation of our study, as most residents follow the indications given by their mentors when dealing with a surgical case. Another limitation is the fact that the answers we received are based on personal opinion and preference in therapy, rather than on objective evidence-based treatment options.

## 6. Conclusions

It is confirmed again that the management of appendicitis is a debatable topic with lack of consensus in many aspects of preoperative, operative, and postoperative care. Discrepancies between the literature and our results were observed, especially regarding the type of surgical approach. Minimally invasive surgery, which has a high support in the available literature regarding safety and usefulness in acute appendicitis in the pediatric population, is not yet widely used by our respondents. In a world with a growing bacterial resistance, an antibiotic stewardship program should exist in each center and further epidemiological studies need to be carried out to have a clearer view of bacterial resistance in each region and establish clinical protocols accordingly.

In conclusion, we recommend expanding the use of minimally invasive techniques when dealing with both complicated and uncomplicated appendicitis. When it comes to the conservative approach, antibiotic resistance from each region should be taken into consideration when choosing anti-biotherapy. We consider that a more homogenous approach to the treatment of acute appendicitis is necessary, which is why clear guidelines regarding the diagnosis and management of appendicitis should be established.

## Figures and Tables

**Table 1 medicina-58-01737-t001:** Number of cases of appendicitis per year, per respondent.

Number of Cases	(*N*, %)
<12	9 (7.6)
12–24	18 (15.3)
25–50	43 (36.4)
51–75	20 (16.9)
76–100	10 (8.5)
>100	18 (15.3)

**Table 2 medicina-58-01737-t002:** Laboratory & imaging tests usage.

	Complete Blood Count (*N*, %)	C Reactive Protein (*N*, %)	Abdominal Ultrasound (*N*, %)	Abdominal CT Scan (*N*, %)	Abdominal IRM Scan (*N*, %)
Never (0%)	0 (0)	1 (0.8)	0 (0)	28 (23.7)	96 (81.4)
Rarely (1–33%)	0 (0)	4 (3.4)	2 (1.7)	82 (69.5)	22 (18.6)
Occasionally (34–66%)	12 (10.2)	16 (13.6)	19 (16.1)	8 (6.8)	0 (0)
Frequently (67–99%)	2 (1.7)	18 (15.3)	24 (20.3)	0 (0)	0 (0)
Always (100%)	104 (88.1)	79 (66.9)	73 (61.9)	0 (0)	0 (0)

**Table 3 medicina-58-01737-t003:** Antibiotics used postoperatively—simple appendicitis.

Antibiotics	*N*, %
Cephalosporins	41 (80.39)
Combinations—cephalosporins and other class(es)	8 (15.69)
Beta-lactams (combinations)—ampicillin/sulbactam; piperacillin/tazobactam	2 (3.92)
Total	51 (100)

**Table 4 medicina-58-01737-t004:** Antibiotics-preoperative use—complicated appendicitis.

Antibiotics	*N*, %
Cephalosporins	63 (57.2)
Combination of cephalosporins and one/more class(es)	33 (30)
Other class(es) or combinations	9 (8.2)
Invalid answers	5 (4.6)
Total	110 (100)

**Table 5 medicina-58-01737-t005:** Antibiotics—postoperative use—complicated appendicitis.

Antibiotics	*N*, %
Cephalosporins	46 (40)
Cephalosporins + Metronidazole	15 (13)
Cephalosporins + Aminoglycosides	14 (12)
Other class(es) and combinations	30 (26)
Invalid answers	10 (9)
Total	115 (100)

**Table 6 medicina-58-01737-t006:** Antibiotics used in appendiceal mass.

Antibiotics	*N*, %
Cephalosporins	36 (38.3)
Cephalosporins + Metronidazole	18 (19.2)
Combination of cephalosporins and one/more class(es) (except Cephalosporins + Metronidazole)	15 (16)
Combination of carbapenems and one/more class(es)	10 (10.6)
Other class(es) or combinations	9 (9.6)
Invalid answers	6 (6.4)
Total	94 (100)

**Table 7 medicina-58-01737-t007:** Interval for appendectomy after conservatively management of appendicular mass.

Interval	*N*, %
6 months	35 (37.2)
3 months	30 (31.9)
6 weeks	13 (13.8)
2 months	10 (10.6)
4 weeks	6 (6.4)
Total	94

**Table 8 medicina-58-01737-t008:** Oral antibiotics at discharge—complicated appendicitis.

Antibiotics	*N*, %
Cephalosporins	52 (68.4)
Beta-lactams (combinations)—amoxicillin/clavulanic acid	6 (7.9)
Invalid answers	6 (7.9)
Other class(es) or combinations	4 (5.3)
Combination of cephalosporins and one/more class(es)	3 (3.9)
Total	76 (100)

## Data Availability

The data presented in this study are available on request from the corresponding author.

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
