# Peer review of "The Management of Pediatric Acute Appendicitis—Survey among Pediatric Surgeons in Romania"

_medicina, 2022, doi:10.3390/medicina58121737_

Round 1
Reviewer 1 Report
it is interesting to see what are the opinions of the surgeons in your country.Some comments: in the methods section is it possible to remove from your percentages the opinion of residents, because they do not decide alone what to do in each patient
also, there are some interesting recent references that you can add about the diagnosis and treatment of appendicitis
i propose one as an example from Di Saveiro et al. in world journal of emergency surgery 2020
have a good luck in your further studies
Author Response
Thank you for your review!
we have addressed some of the language mistakes that could be found in the previous draft.
We have also updated our references.
Reviewer 2 Report
Since appendicitis is common and not difficult to treat, each hospital treats it with various treatment strategies.
This paper is a national-wide questionnaire survey conducted by the Romanian Society of Pediatric Surgery, and is useful for understanding the current state of treatment of appendicitis in Romania, and for determining treatment policies in countries similar to Romania. Although no new medical knowledge was discovered, it is important and interesting to clarify the treatment situation of appendicitis in one country.
There is no point to be revised.
Author Response
Thank you for your review!
Reviewer 3 Report
The authors assessed the current practice pattern in the management of pediatric acute appendicitis in Romania.
They concluded that there seems to be a lack of consensus regarding several aspects of the management of acute appendicitis in children. In addition, minimally invasive surgery is not as widely used as reported, despite of the literature support.
My concerns are as follows:
1. Introduction The authors did not mentioned appendicitis scores in introduction. As they used PAS / Alvarado score in their questionnaire this should mentioned and adequate reference should be included. As the majority of the center recently use Appendicitis Inflammatory Score (AIR) the authors should emphasize that acute appendicitis can be detected with a high level of sensitivity and specificity using the AIR score. Also, the AIR score may differentiate perforated from non-perforated appendicitis in patients with a high level of accuracy. REFERENCE: Validity of appendicitis inflammatory response score in distinguishing perforated from non-perforated appendicitis in children. Children (Basel). 2021;8(4):309. doi: 10.3390/children8040309.
2. Introduction – Introduction is also insufficient in regards to biomarkers used for the diagnosis of acute appendicitis. Several markers such as sodium, bilirubin, RDW, etc. have recently been extensively investigated. Please include several sentences regarding laboratory markers and following references:
Hyponatremia as a predictor of perforated acute appendicitis in pediatric population: A prospective study. J Pediatr Surg. 2021;56(10):1816-1821. doi: 10.1016/j.jpedsurg.2020.09.066.
Hyperbilirubinemia as an Indicator of Perforated Acute Appendicitis in Pediatric Population: A Prospective Study. Surg Infect (Larchmt). 2021;22(10):1064-1071. doi: 10.1089/sur.2021.107.
Anand S, et al. Utility of red cell distribution width (RDW) as a noninvasive biomarker for the diagnosis of acute appendicitis: A systematic review and meta-analysis of 5222 cases. Diagnostics (Basel). 2022;12(4):1011. doi: 10.3390/diagnostics12041011
3. Design of the study – The main problem of this study is its design. Methodology is very poorly described. The authors used questionnaire which they designed but I do not see that the validation of the questionnaire was performed. This is mandatory for such type of the study.
4. Sample size calculation – The sample size calculation is one of the most important factors which influence the quality of the study. There is no sample size calculation in methodology, which is one of the most important study design parameters. This should be described in a paragraph regarding statistics. What was the power of the calculated sample? Next how the authors choose sample, random sample may be (and probably it is) major source of bias. They should calculate proportion of general hospitals, rural hospitals, university hospital, than experience of the surgeons… and then calculate how many questionnaires is required for each group I do not see that any of that was performed.
5. Methodology – Primary and secondary outcomes of the study should be clearly stated in methodology. Please update.
6. Quality of English should be improved. Manuscript should be edited for the language from a native English speaker or professional language editing service to improve the grammar and readability.
7. Limitations of the study – There are many limitations of the study that were not even been mentioned. I think the authors should seek help from expert in regards to many parts of this investigation.
8. Finally, the main question arises: What is the importance of this study and what this study adds to the literature? Treatment strategies of acute appendicitis are well-known and reported several times in literature. This may be appropriate for local journal but not international such as Medicine.
Author Response
Dear reviewer,
Thank you for your comments.
In the introduction we mentioned as suggested the scores used for the diagnosis of acute appendicitis and we also touched upon predictors such as hiponatremia, hyperbilirubinemia and RDW.
We also modified the methodology in order to hopefully give a clearer view of our intensions with this study. And we also mentioned our reasoning regarding sample size calculation.
We tried to correct all language errors.
We also added something more to the limitations of our work.
We are looking forward to hearing your suggestions.
Thank you again
Round 2
Reviewer 1 Report
thank you very much for your answers. However you don’t answer about the residents in the study if it is not possible to remove them from your be study!!! You must write something about this in the limitations because I think residents don’t decide alone.
Author Response
Thank you for your comment. We added a new limitation to the study regarding the answers we received from the residents., as per your suggestion.
Reviewer 3 Report
Although the authors made efforts to revise the study, this study is still poorly designed and has significant problems with methodology and study design. The main criticisms remain the nonvalidated questionnaire, inadequate inclusion of surgeons in the study (rural/urban hospitals), and lack of novelty. I do not believe that this study has scientific value to be published in Medicina.
Author Response
Thank you for your response. The questionnaire was valided by the members of the scientific committee of our society. In our country we have no rural hospitals that have pediatric surgery departments. All pediatric surgeons work in urban hospital of different sizes, of course, including university hospitals. If pediatric patients present to a rural hospital they are referred to the emergency departament of the first urban hospital with a pediatric surgery department.